Soil organic nitrogen variation shaped by diverse agroecosystems in a typical karst area: evidence from isotopic geochemistry

Han Ruiyin 1 2
Zhang Qian 3
Xu Zhifang 1 2 4 zfxu@mail.iggcas.ac.cn
1 Institute of Geology and Geophysics, Chinese Academy of Sciences , Beijing , China
2 University of Chinese Academy of Sciences , Beijing , China
3 Institute of Geographic Sciences and Natural Resources Research , Beijing , China
4 CAS Center for Excellence in Life and Paleoenvironment , Beijing , China
Shahzad Tanvir
Electronic publication date: 2024 Apr 15
Publication date: 2024
Volume: 12
Electronic Location ID: e17221
Received 2023 Nov 7; Accepted 2024 Mar 20
Copyright: © 2024 Han et al.
Copyright year: 2024
Copyright holder: Han et al.
License: This is an open access article distributed under the terms of the Creative Commons Attribution License, which permits unrestricted use, distribution, reproduction and adaptation in any medium and for any purpose provided that it is properly attributed. For attribution, the original author(s), title, publication source (PeerJ) and either DOI or URL of the article must be cited.
License URL: https://creativecommons.org/licenses/by/4.0/

Keywords: δ15NSON, Nitrogen cycling, Land management, Agricultural disturbance, Karst area

Funding: National Key Research and Development Program of China 2020YFA0607700 National Natural Science Foundation of China 41730857, 42273050 and 42203011 Key Research Program of the Institute of Geology & Geophysics, CAS IGGCAS-202204 Youth Innovation Promotion Association CAS 2019067 This research was financially supported by the National Key Research and Development Program of China (grant No. 2020YFA0607700), the National Natural Science Foundation of China (grant Nos. 41730857, 42273050, 42203011), the Key Research Program of the Institute of Geology & Geophysics, CAS (grant No. IGGCAS-202204), and the Youth Innovation Promotion Association CAS (2019067). The funders had no role in study design, data collection and analysis, decision to publish, or preparation of the manuscript.

==============================
Background

Soil organic nitrogen (SON) levels can respond effectively to crop metabolism and are directly related to soil productivity. However, simultaneous comparisons of SON dynamics using isotopic tracing in diverse agroecosystems are lacking, especially in karst areas with fragile ecology.

Methods

To better understand the response of SON dynamics to environmental changes under the coupling of natural and anthropogenic disturbances, SON contents and their stable N isotope (δ15NSON) compositions were determined in abandoned cropland (AC, n = 16), grazing shrubland (GS, n = 11), and secondary forest land (SF, n = 20) from a typical karst area in southwest China.

Results

The SON contents in the SF (mean: 0.09%) and AC (mean: 0.10%) profiles were obviously lower than those in the GS profile (mean: 0.31%). The δ15NSON values ranged from 4.35‰–7.59‰, 3.79‰–7.23‰, and 1.87‰–7.08‰ for the SF, AC, and GS profiles, respectively. Decomposition of organic matter controlled the SON variations in the secondary forest land by the covered vegetation, and that in the grazing shrubland by goat excreta. δ15NSON ranges were controlled by the covered vegetation, and the δ15NSON fractionations during SON transformation were influenced by microorganisms in all surface soil.

Conclusions

The excreta of goats that contained 15N-enriched SON induced a heavier δ15NSON composition in the grazed shrubland. Long-term cultivation consumes SON, whereas moderate grazing increases SON content to reduce the risk of soil degradation. This study suggests that optimized crop-livestock production may benefit the sustainable development of agroecosystems in karst regions.

Introduction

As an indispensable media for nutrient cycling, soil is recognized as a functional pool for biogeochemical processes of nutrient components in terrestrial ecosystems. Soil nutrients play a crucial role not only in regulating soil productivity and crop production but also in influencing global climate change (Harris et al., 2022; Saiki et al., 2020). The excess release of nitrous oxide from nitrogen (N) cycling in the soil is also detrimental to tropospheric ozone (Park et al., 2012; Russenes et al., 2019). Nitrogen oxide is a typical long-lived greenhouse gas that is mainly produced by the soil microbiota (Harris et al., 2022). Additionally, the dynamics of N may constrain C stocks in terrestrial ecosystems (Sisti et al., 2004). Release from industrial fuel burning deteriorates climate change, and anthropogenic inputs (e.g., inorganic/organic N fertilization) also strongly influence the balance of N in ecological systems (Choi et al., 2017; Kimetu et al., 2004). Generally, soil organic nitrogen (SON) occupies a large proportion (>90%) of the soil total nitrogen, which is mainly supported by N mineralization and external inputs (Li et al., 2018). More than half of the available N assimilated by plants (mainly applied by inorganic N) is released by the mineralization of SON (Luce et al., 2014). The primary intruders of SON distribution are attributed to the application of organic N fertilizer and excretion of livestock during grazing under agricultural utilization (Li et al., 2022). Moreover, over-dose inputs of N may also have a risk of N losses in soils and further enhance the imbalance of N in the whole ecological system (Yu et al., 2019). SON loss can also indicate soil erosion and can reduce multiple soil functions that contribute to land degradation (Qiu et al., 2021). Therefore, clarification of SON dynamics and the mechanism of SON stability are necessary and may provide an important foundation for predicting climate projections and land evolution.

The transformation processes of nutrients can be effectively clarified by isotopic tracing in ecosystems (Liu & Han, 2021a; Qu & Han, 2022, 2023). The stable N isotope (i.e., δ15NSON) has also been widely applied to explore N cycling in terrestrial ecosystems (Liu et al., 2022b; Yu et al., 2019; Zeng et al., 2023). Biochemical activities (such as microbiota utilization, plant assimilation, and animal excretion) are the primary factors that induce the large fractionation of δ15NSON during the transformation of N (Liu, Han & Li, 2021; Liu et al., 2022a; Luce et al., 2014; Zhang et al., 2022). Animal are usually enriched 15N in the tissues and prefer to excrete 14N (He et al., 2015). The synthetic fertilizer generally carries out abundant 15N-depleted SON (δ15NSON = −0.5 ± 2.5‰), while more than 90% raw manure (i.e., animal excreta) carries out 15N-enriched SON (δ15NSON = 8.5 ± 5.5‰) (Choi et al., 2017). Hence, the δ15NSON composition of the soil environment may be strongly governed by agricultural activities. Previous studies in the field of N cycling have mainly focused on the storage of individual forms of N (such as NH4+-N and NO3–-N) or the distribution of soil total N, as well as the shallow soil of a specific type during long-term monitoring (Bilotto et al., 2022; Xiao et al., 2018). Comparative analyses of SON variation and δ15NSON composition are limited in whole soil profiles under different land utilizations, especially in karst areas. Understanding regional N biogeochemical cycling requires analysis of δ15NSON fractionation and influencing factors, which can improve soil stability and optimize land utilized management.

In the karst region of southwest China, the soil environment is characterized by high ecological fragility and a high risk of soil erosion (Chang et al., 2018; Zhang et al., 2019). The widespread distribution of calcareous soils with lower thickness is highly susceptible to irreversible environmental damage (e.g., soil nutrient loss, land degradation, and rocky desertification) owing to a slow recovery rate (Chang et al., 2018; Liu, Han & Zhang, 2020). The use of N fertilizer and sewage discharge may also aggravate soil degradation. However, agriculture is the major industry in most areas of Guizhou Province, and is considered an initiator to decrease the storage of soil nutrients. Studies in the field of the N cycling have focused on the discussion of N levels on a large scale or on the dynamics of SON contents only in surface-cultivated soils (Xiao et al., 2018; Zhang et al., 2022). Insufficient attention has been paid to the entire soil profile under various land uses in a small catchment with a similar geological background, especially for karst areas with high ecological sensitivity. The information carried by a suite of the soil profile can comprehensively interpret how disturbances affect the N cycling in the sampling region. Moreover, the analysis of N under diverse land uses can also evaluate soil evolution in a specific region (e.g., abandoned cropland–shrubland–forestland). The implications of the N cycling in soil profiles under diverse land-use types deserve further exploration. Yinjiang County is a typical karst area in Guizhou Province, Southwest China, and its economy is mainly supported by agriculture. However, the high risk of soil erosion and nutrient loss is widely observed in karst areas, which limits production. This study focused on karst soil profiles under different land utilizations in Yinjiang County to (1) determine the spatial patterns of SON contents and δ15NSON compositions under diverse agricultural regulation modes, (2) explore the driving factors influencing soil quality by δ15NSON isotopic tracing, and (3) identify the alteration mechanisms of diverse agricultural disturbances on SON cycling. This study provides a practical basis for optimizing agricultural management and subsequent recovery to sustainably improve soil availability in karst areas.

Materials and Methods

Study area description and soil sampling

The study area (27°35′–28°21′N and 108°18′–108°48′E, Fig. 1) is situated in Muhuang Town, the largest town in Yinjiang County, which is a karst area in northeast Guizhou Province. Yinjiang County possesses 13 towns with a 1,968.1 km2 area, and the average population intensity is 145 persons per square kilometer. Yinjiang County is controlled by a subtropical monsoon climate with a combination of precipitation and heat. The number of rainfall days in 2016 was 147, with the majority occurring in summer, and the annual relative humidity was 78% in Yinjiang County (Tongren Bureau of Statistics, 2017). The extent of soil erosion in southwestern China is shown in Fig. 1A. Most soils in Yinjiang County slowly evolved from carbonate rocks and sandstone, which are characterized by weak soil aggregate stability and strong leaching (Zhang et al., 2021). Furthermore, the abundant mountains and frequent high-intensity precipitation in the study area also easily induce soil loss. The statistical yearbook of Tongren City in 2017 reported that the actual arable land covered 18,588 hectares and crop production reached 144,800 tons, while the stock of goats reached 66,100 at the end of 2016 in Yinjiang County (Tongren Bureau of Statistics, 2017).

Figure 1 Relief degree of land surface (RDLS, A), lithology (B), and land utilizations (C) of three sampling profiles in the study area.

The data set was provided by the geographic remote sensing ecological network platform (http://www.gisrs.cn/).

A large amount of cropland has been abandoned to protect the ecological environment under the policy of grain for green in the last 20 years. The afforestation area reached 9,171.3 hectares in 2016 (Tongren Bureau of Statistics, 2017). However, the total area of arable land has limited variation, and the number of grazing goats has increased in recent years because agriculture is considered the primary industry in Yinjiang County. The wide distribution of abandoned cropland and forest land provides an appropriate opportunity to analyze the disturbance of soil properties by agricultural activities. A total of 47 samples were collected from three soil profiles under different land uses in September 2016 (Fig. 1). Of which 20 soils were obtained in the profile under secondary forest land (SF profile), 16 soils were obtained in the profile under abandoned cropland for 3 years (AC profile), and 11 soils were obtained in the profile under grazing shrubland for 5 years (GS profile). This study represents multiple stages of abandoned cropland evolution using three land uses (abandoned cropland–shrubland–forest land). Long-term changes can also be employed as an alternative by substituting the spatial variation for reference with temporal changes within a given region (Xiao et al., 2018). The detailed description of sampling methods was reported by Han & Xu (2022), more than three profiles were dug within a distance of 1 m at each site. All results shown in this study are averages of samples from duplicate profiles at the same depth. An interval of 5 cm was applied to the sampled surface soil, and an interval of 10 cm was applied to the deeper soil. Based on the depth from the surface to the bedrock, the thicknesses of the SF and GS profiles were 160 and 70 cm, respectively. According to the soil sampling standards in China, the sampling depth in croplands was defined as 100 cm. Therefore, a sampling depth of 130 cm was selected to ensure data reliability in the AC profile.

Determination of soil composition

All chemical analyses were based on dried soil samples and excluded plant residues and stones. The soil texture was determined using a laser particle size analyzer, and a soil particle diameter less than 2 μm was assigned to clay, ranging from 2 to 53 μm in the silt, and from 53 to 250 μm in the sand (Soil Survey Staff, 2014). Soil pH was measured by a pH meter in the suspension of soil (sample diameter < 2 mm): water = 2 : 5 with a precision of ± 0.05 (Han et al., 2020). The remaining samples were sieved at 75 μm (approximately 200 mesh) for composition measurements. Finer soils were digested with 2 mol/L KCl for 1 d to remove inorganic N and dried at 55 °C after repeated washing to neutral pH with pure water. SON and SOC contents were measured using treated soil samples with an elemental analyzer (Vario TOC cube; Elementar Analysensysteme GmbH, Langenselbold, Germany) with a precision of ±0.02%. The SOC contents and δ13CSOC values have been reported by Han, Zhang & Xu (2023a). The nitrogen isotopes were determined using a stable isotope mass spectrometer (Thermo, MAT-253, USA) with a precision better than 0.2‰. All values of δ15NSON (15NSON/14NSON) were normalized to the atmospheric N2 standard in ‰ unit:

δ15NSON=[(δ15NM−δ15NA)/δ15NA]×1000

where M means measured samples, and A means atmospheric N2 standard. All measurements were performed at the Institute of Geographic Sciences and Natural Resources Research, CAS, with duplicate determination to ensure accuracy.

The map of the study area was graphed using ArcMap (version 10.8; Esri ArcGIS Desktop; Esri, Redlands, CA, USA). All data analyses of SON content, δ15NSON values, and related soil properties were performed using SPSS (version 25.0; IBM SPSS Statistics, Chicago, IL, USA), and all figures were analyzed using Origin (version 2017; OriginLab, Northampton, MA, USA).

Results

Soil properties

The data of the soil physical parameters are given in Table S1, and the soil particle distributions in the three profiles are plotted in Fig. 2. Detailed data on soil pH variation and soil particle distributions have been reported by Han, Zhang & Xu (2023b). The AC profile had an acidic soil environment, and medium to slightly alkaline soil environments were observed in the SF and GS profiles. The three soil profiles in Yinjiang County were characterized by lower clay and sand contents, according to the United States Department of Agriculture (Soil Survey Staff, 2014), and the soils were silt and silt loam soil types. Stronger soil coarsening in abandoned croplands showed higher desertification threats than in other lands. A larger proportion of sand fraction was observed in the AC profile, and soil particle distributions were more variable in the AC profile.

Figure 2 Soil texture diagram of the three profiles.

Geochemical characteristics

Soil organic carbon (SOC) has been suggested to be an essential factor affecting SON dynamics by regulating N release from plants (Han et al., 2020). The variations in SON content, δ15NSON values, and the ratios between soil organic carbon (SOC) content and SON content (SOC/SON, all values in this study are mass ratios) under the three land uses in Yinjiang County are listed in Table S2, and the ranges and coefficient of variations (CV) are presented in Table 1. The SON contents ranged from 0.06% to 0.23%, 0.08% to 0.22%, and 0.12% to 0.63% in the SF, AC, and GS profiles, respectively. The values of δ15NSON ranged from 4.35‰ to 7.59‰ in the SF profile, 3.79‰ to 7.23‰ in the AC profile, 1.87‰ to 7.08‰ in the GS profile. The varied range of SON contents in the three soil profiles (0.06% to 0.63%) were similar to those in the Puding catchment (0.05% to 0.77%) and Libo County (0.08% to 0.75%) with karst structure, and lower than that in the Jiulongjiang Basin (0.16% to 0.91%), while the δ15NSON compositions in the study area (1.11‰ to 9.46‰) were narrower than those in the Puding catchment (1.87‰ to 7.59‰) and Libo County (1.20‰ to 10.0‰), but obviously higher than that in Jiulongjiang Basin (0.8‰ to 5.7‰) (Han et al., 2020; Li, Han & Tang, 2014; Liu & Han, 2022). The ratios of [SOC/SON] ranged from 5.21 to 18.13 in the SF profile, 4.81 to 8.15 in the AC profile, and 6.11 to 17.18 in the GS profile. The CV values showed vertical variations in SON contents, and its isotopic fractionations were limited in the AC profile and animated in the GS profile.

Table 1 Values of SON, δ15NSON, and [SOC/SON] in the three karst soil profiles under diverse land uses.

Sampling sites	Parameters	SON (%)	δ15NSON (‰)	SOCa/SON	
SF profile	Mean	0.09	6.83	8.88	
Min	0.06	4.35	5.21	
Max	0.23	7.59	18.13	
SD	0.04	0.76	3.24	
CV	42.74%	11.08%	36.48%	
AC profile	Mean	0.10	6.05	6.76	
Min	0.08	3.79	4.81	
Max	0.22	7.23	8.15	
SD	0.03	0.88	0.75	
CV	32.93%	14.61%	11.09%	
GS profile	Mean	0.31	4.78	9.66	
Min	0.12	1.87	6.11	
Max	0.63	7.08	17.18	
SD	0.19	1.80	3.27	
CV	63.74%	37.72%	33.82%	
Note:

a SOC contents have been reported by Han, Zhang & Xu (2023a).

The vertical distributions showed that the SON contents, δ15NSON values, and ratios of [SOC/SON] fluctuated greatly within the 0 to 30 cm soils of the three profiles (Fig. 3). SON content was in the sequence of GS profile (mean: 0.31%) > SF profile (mean: 0.09%) ≈ AC profile (mean: 0.10%) and tended to a similar value in the soils below 30 cm. In contrast, the δ15NSON values were at the highest level in the AC profile and the lowest level in the GS profile in the 0 to 30 cm soil layer at the same depth. The values of δ15NSON increased in the 0 to 30 cm soils of the three profiles. Notably, the δ15NSON values consistently increased in soils below 30 cm in the GS profile, whereas those values hardly changed in the SF and AC profiles. The variations in SOC/SON ratios decreased as the soil depth increased in the SL (range 9.65) and GS profiles (range 8.82), especially in the soil above 30 cm. The values of SOC/SON remained fairly constant in the AC profile (range 3.35) and were almost at the lowest level among the three profiles.

Figure 3 Vertical variation of (A) SON contents, (B) δ15NSON values, (C) ratios of SOC and SON in the three profiles.

Discussion

Spatial patterns of SON contents under diverse disturbed mode

Land evolution and utilization possibly more affect the variation in SON contents, rather than the climate and geological background because of their analog within the small catchment. The spatial patterns of SON in the three profiles were presented in Fig. 3. Covered vegetation can influence the contents of SON and the bioaccumulation of soil microbes by regulating litter inputs (Ambrosino et al., 2021; Yan et al., 2020). Furthermore, the SON contents are strongly controlled by microbiotas activities by N mineralization, ammonium, nitrification immobilization, and nitrate immobilization (Buzin et al., 2019; Li et al., 2018). The microbial biomass is generally at the highest level in forests and at the lowest level in agricultural land (Li et al., 2018). However, a lower SON content was observed in the AC profile. It may be closely linked to the application of manure and nitrogenous fertilizers, which are used to promote crop growth in agricultural activities. Synthetic N fertilizer is generally considered the largest contributor to the N level of cultivated soil, which is also accompanied by high loss rates of N (such as NO3- leaching and NO2 emission) (Choi et al., 2017). However, SON will continue to decompose and be assimilated by plants to become lost in abandoned croplands with the cessation of the N fertilizer supply (Liu et al., 2010). In addition to the utilization of inorganic N by plants, soil organism is also regarded as forceful factor in regulating SON dynamics. The distribution of soil microorganisms plays a forceful role in shaping the SON pool size by facilitating the conversion of inorganic to organic N and serves as a crucial repository (Buzin et al., 2019). The biomass accumulation of microbiotas tends to be higher in shrubs than in forests or areas without plants, particularly in the surface soil (Yan et al., 2020). Microbiotas may cause the highest level of SON contents in the surface soil of the GS profile.

The SOC/SON ratios in the three profiles are shown in Fig. 3. The variation in the SOC/SON ratios in the cultivated land (<8.5) was significantly lower than that in the other lands, while the SON content in the cultivated land was similar to that in the forest land. It possibly has resulted from lower levels of SOC contents (0.51% to 1.77%) in the AC profile (Han, Zhang & Xu, 2023a). Following the abandonment of cultivation, the cessation of fertilizer supplies disrupted soil components to lose SOC and could not revert to the original SOC levels within a short period. The SON contents in the GS profile were much higher than that in the SF profile, whereas the SOC/SON ratios showed similar values, indicating the existence of additional SOC and SON in the GS profile. Neither forested nor abandoned land had additional organic matter input in recent years, while goat-grazed shrubland land may sustainably obtain soil organic matter (SOM) from goat excreta, which may cause the obviously higher SON content in the surface soils of the GS profile. The SOC/SON ratios presented obvious decrease trends in the 0 to 30 cm soil layers in both the SL and GS profiles. Generally, SOM is enriched in surface soil and decreases with increasing depth, which is regarded as a forceful factor in regulating the dynamics of SON (Liu, Han & Zhang, 2020). Higher ratios of SOC/SON were less suitable for SOM decomposition than lower ratios of SOC/SON because of restraining by microorganisms (Lan, Hu & Fu, 2020). The higher SOC/SON in the SF and GS profiles may have resulted from additional SOC inputs by plant litter and excreta. The accumulation of SOM in deeper soils mainly combines with Ca2+ to form stable complexes, especially in calcareous soils (Li et al., 2018). The lower SOC/SON ratios with limited fluctuation in the deeper soil of the three profiles may indicate a stable form of SOM, which was difficult to decompose.

Environmental factors on SON contents and isotopic composition

Soil texture can also regulate SON and δ15NSON isotopic fractionation in soil. Soil texture can directly influence SON retention and storage, which is easily altered by land uses conversion (Lan, 2021; Xia, Rufty & Shi, 2020). Additionally, increasing proportions of silt and clay can also control the δ15NSON composition by increasing the abundance of microbes (such as fungi and filamentous bacteria) (Hobbie & Ouimette, 2009; Xia, Rufty & Shi, 2020). In this study, the δ15NSON values were stable in the deep soils of all profiles, which may inherit from the characteristics of regional bedrock and have been less disturbed of biological activities. However, the soil components are easily transferred during pedogenesis. The relationships among SON content, δ15NSON value, Ca content, soil pH, and soil particle proportion are illustrated in Fig. 4. Both soil pH and Ca content showed no correlation with SON content and δ15NSON value in the SF profile, suggesting that the controlling factor of SON dynamics could be attributed to others. Soil pH was negatively correlated with SON content in the GS profile but positively correlated with δ15NSON values in the AC and GS profiles. Ca content showed positive correlations with SON content and a negative correlation with δ15NSON values in both the AC and GS profiles. The complexions between organic matter and minerals are strongly affected by soil Ca, and polyvalent cations (the primary cations are Ca and Mg in neutral to alkaline environments) can promote SOM stability (Li et al., 2017). The Ca content in the study area is at a higher level in Chinese soil because of the karst structure (China Environmental Monitoring Station (CEMS), 1990). The complexation of the OM-Ca2+-minerals is considered a vital mechanism to reserve SOM in karst areas in southwest China due to the enrichment of Ca (Li et al., 2017; Liu & Han, 2021b). Therefore, Ca may play a vital role in SON distribution by increasing the stability of the remaining SON in the study area. The fraction of SON fixed by minerals was possibly enriched with lighter N isotopes in the AC and GS profiles.

Figure 4 Pearson correlation coefficients among SON contents, δ15NSON values, and other soil characteristics in the SF profile (A), AC profile (B), and GS profile (C).

A positive correlation was found between the SON content and sand proportion in the SF profile, and negative correlations were found between the clay proportion and SON content in the AC and GS profiles. The N requirement for abundant biomass and rapid turnover of microbiotas in the shrubland may be responsible for accelerating SON decomposition (Stoner et al., 2021; Xia, Rufty & Shi, 2020). In the sand fraction, the enzyme-mediated biochemical mineralization of SON is stronger and the SON content decreases, whereas clay supplies more capacity for SON by the stabilization of organic-mineral compounds (Xia, Rufty & Shi, 2020). The results showed that all soil profiles were silt loamy in texture, and the sand proportions were at the highest level in the AC profile (4.00% to 9.38%, sand proportions in the other profiles ranged from 0.52% to 2.66%), which may improve the loss of SON. Generally, soil properties can indirectly affect δ15NSON fractionation by controlling the stability and diversity of microbial biomass and products (Khalil, Hossain & Schmidhalter, 2005). Only the sand fraction showed a correlation with the SON content and δ15NSON value, whereas the proportion of sand was less than 1.5% in the SF profile. It can be assumed that the influence of the soil texture on δ15NSON is limited. The δ15NSON values were negatively correlated with the clay and silt fractions in the AC and GS profiles, which is consistent with the silt loamy soil. Therefore, 15N-enriched N may prefer to remain in clay and silt particles.

SOC and δ13CSOC were employed to further discuss the factors controlling the transformation of SOM (Fig. 5). Except for the AC profile, the SOC/SON ratios showed a positive linear correlation with the SON contents in the other profiles. It can be suggested that SOM decomposition strongly influenced the distribution of SON in the SF and GS profiles. The lower SOC and SON contents in the AC profile may have resulted from elimination by leaching and crop uptake. Negative correlations between SON contents and δ15NSON values were also found in the three profiles (Fig. 5B), which suggests that the transformation of SON affected δ15NSON fractionation. Generally, the decomposition rate of SOM increases when the mass ratio of SOC/SON is less than 25, and at a higher level when the ratio is less than 15 (Brust, 2019). The SOC/SON mass ratios in almost all soils were less than 10, indicating a high rate of SOM decomposition. The 15N-enriched N prefers to accumulate in microorganisms and nitrifying bacteria, and returns to the soil in the form of residual SOM by dead microbiota (Craine et al., 2015). In contrast, 15N-depleted N is produced in the soil by SON mineralization and nitrification, and is subsequently lost or absorbed by plants (Craine et al., 2015; Li et al., 2018). Therefore, the SOC/SON values were negative for δ15NSON in most soils of the three profiles (Fig. 5C). The correlation between SOC/SON and δ15NSON values was slightly positive in the SF profile below 20 cm, with a narrow variation in δ15NSON values (1.14‰), which possibly indicated the limited influence of SOM decomposition.

Figure 5 Correlation among SON contents and δ15NSON values, [SOC/SON] ratios, δ15NSON values, and δ13CSOC values (data from Han, Zhang & Xu (2023a)) in the three soil profiles.

The full line means p < 0.01, and the dashed line means p > 0.05.

Soil δ15NSON fractionation affected by agricultural activities

As an active area for plant–soil–microbiota interaction, the surface soil layer is a container, in which a large amount of material transformation and energy exchange occurs (Daly et al., 2021). Fluctuations in δ15NSON values were also concentrated in the soils over 30 cm in the three profiles (Fig. 3). In addition to direct inheritance from direct SON contributors (e.g., fertilizer and plant litter), the δ15NSON composition also changes during the transformation of various N compounds and biological processes (Han et al., 2020; Loss et al., 2017). Generally, plants prefer to assimilate 15N-depleted N after SON nitrification, whereas heavier δ15NSON is enriched in microbiota (Dijkstra et al., 2006; Hobbie & Högberg, 2012). Therefore, the 14N will be removed, and the 15N will be accumulated in deeper soil. Due to the decomposition of SOM, the δ15NSON and δ13CSOC values generally increase downward in natural land without the transition of C3 to C4 covered vegetation and exogenous disturbance (Han et al., 2020). The vertical δ15NSON composition in the three profiles was possibly controlled by the plants. Analyses of δ15NSON and δ13CSOC were employed to further analyze nutrient cycling in plant–soil systems (Fig. 5D). C3 plants widely covered by vegetation in the study area may have a great influence on the isotopic composition of soils. The δ15NSON values of leguminous plants (mean δ15NSON: 1‰) and non-leguminous plants (mean δ15NSON: 9‰) have a large difference (Deniro, 1987; Hoefs, 2009). Obviously, the distribution of δ15NSON values trended toward the end-member of non-leguminous plants in the GS profile while the δ15NSON composition of soils in the SF and AC profiles may be strongly influenced by leguminous plants. Plant uptake and utilization may be directly affected by changes in the soil environment caused by agricultural disturbances. Prolonged fertilizer input during tillage easily causes a decrease in δ15NSON values in the soil due to the relatively lower value of δ15NSON (Bateman & Kelly, 2007). The long-term cultivation of the AC profile location was accompanied by abundant inputs of synthetic N fertilizer; thus, the δ15NSON values in the AC profile were relatively lower. In contrast, the decomposition of organisms may induce sustainable accumulation of 15N-enriched materials, which can be improved during the tillage process (the largest reach within 5–7‰) (Hoefs, 2009). Therefore, the δ15NSON values in the deeper soil were higher than those in the shallow layers of cropland.

However, the δ15NSON values were lower in the surface soils of the GS profile than in those of the others. Létolle (2012) reported that δ15N values in soils mostly varied from 5.1‰ to 12.3‰. However, the values of δ15NSON in the topsoil were lower than 5.1‰, particularly in the GS profile (δ15NSON < 5.1‰ in soils at 0–25 cm depth). Previous studies have demonstrated that a higher grazing intensity would lead to less N residence time in the soil, and further decrease the δ15NSON value (Golluscio et al., 2009). Additional SOM inputs may promote ammonification, and further lead to an increased ratio of SOC/SON and soil pH (Khalil, Hossain & Schmidhalter, 2005). Similar to the unreturnable crops in the cropland, returnable plant biomass on the grazing shrubland was reduced. The N ingested by animals is absorbed and transformed, and lighter N is generally excreted preferentially, while heavier N is enriched within the tissues (Wittmer et al., 2011). Generally, the N-isotopic composition will be heavier in animals than in ingested materials, especially in elderly individuals (Wittmer et al., 2011). Goat grazing may be an important factor causing 15N depletion in the SON of the GS profile. Moreover, the rate of SON mineralization in grazed land is lower than that in non-grazed land (Golluscio et al., 2009). Therefore, δ15NSON values were lower in the GS profile than in the other profiles.

Erosion risk of soil nutrient loss in karst area

Long-term application of organic fertilizer regulates soil SON and SOC levels, while adversely affecting the stability of SON (Xu et al., 2023). Unreasonable agricultural structures possibly cause soil acidification, which also accelerates the loss of soil nutrients by inhibiting the transformation of SON to available N in plants (Brust, 2019). Soil acidification also promotes NO3– leaching and the loss of exchangeable cations (e.g., Ca and Mg) (Čakmak et al., 2014). Moreover, it also increases the solubility of potentially toxic heavy metals in the soil and ultimately threatens human health (Čakmak et al., 2014). Different land uses can alter SON dynamics by changing the soil biogeochemistry under artificial disturbance (Fig. 6). The pH values ranged from 4.8 to 5.2 in the AC profile, showing an obvious acidification trend. The Fe, Mn, and Pb contents in the abandoned cropland were also higher than those in shrubland, while both were lower than those in the forest land in Yinjiang County (Han & Xu, 2022; Han, Zhang & Xu, 2023b). It confirms that cultivation may cause damage to soil quality and cannot be recovered within a short time after the cessation of agricultural activities. As shown above, sustainable cultivation results in relatively large soil disturbances, and self-recovery after stopping tillage makes it difficult to restore soil quality in the short term. In contrast, the SON content of shrublands possessed a higher level to sustain soil N supplementation.

Figure 6 The SON cycling under different land uses in the Karst area.

The karst area in Guizhou Province is characterized by abundant precipitation and predominantly mountainous areas that are sensitive to environmental changes. A high soil moisture content may promote the conversion of SON to inorganic N, resulting in the loss of SON (Li et al., 2022). Additionally, a large slope further intensifies the loss of soil nutrients by soil erosion. Furthermore, the additional N may also stimulate the emission of N2O, and be further unfavorable to the eco-environment (Harris et al., 2022). However, slope farming is widely distributed in the Guizhou Province. Agriculture has become the primary economic industry in many parts of the Guizhou Province. Crop–livestock production systems have been demonstrated to improve the utilization of organic fertilizer (i.e., animal excreta) rather than chemical fertilizer and soil stability, further forming regional nutrient recycling (Jalilpour, Chavoshi & Jalalian, 2022; Xu et al., 2023). Combining crop–livestock production with agriculture may reduce the degradation of fragile lands compared with the simple type. It is important to focus on changes in soil quality and soil nutrient loss, and to optimize agroecosystem functions in the karst region of southwest China.

Conclusions

SON transportation and δ15NSON composition were greatly controlled by covered vegetation and agricultural disturbances in Yinjiang County. Plant debris contributes abundant SOM in the secondary forest land, while crop production picking and goat assimilation in the shrubland prevent the return of plant litter. Moreover, fertilizer input during cultivation changed the nutrient supply and soil characteristics to improve the rate of SON losses. Long-term cultivated soil easily leads to greater loss of nutrients under unprotected and challenging conditions to recover the ecological balance by self-restoring in a short period, especially in silt loamy soils with a poor capacity for N storage. Moderate grazing may be beneficial for maintaining soil fertility in karst soils, whereas the recovery management of cropland after abandonment should be considered, especially for SON retention. The transition from the traditional cultivation form to the crop–livestock production system should be considerably adjusted in karst regions to prevent soil degradation and support sustainable productivity.

Supplemental Information

Supplemental Information 1 Soil profiles description and basic characteristics in the study area.

Supplemental Information 2 SON contents, δ15NSON compositions, ratios of SOC/SON in the Yinjiang County.

The authors thank Dr. Yang Tang from the Institute of Geochemistry, Chinese Academy of Sciences for field sampling.

Additional Information and Declarations

Competing Interests

Author Contributions

Data Availability

The authors declare that they have no competing interests.

Ruiyin Han conceived and designed the experiments, performed the experiments, analyzed the data, prepared figures and/or tables, authored or reviewed drafts of the article, and approved the final draft.

Qian Zhang conceived and designed the experiments, performed the experiments, prepared figures and/or tables, authored or reviewed drafts of the article, and approved the final draft.

Zhifang Xu conceived and designed the experiments, analyzed the data, prepared figures and/or tables, authored or reviewed drafts of the article, and approved the final draft.

The following information was supplied regarding data availability:

The raw measurements are available in the Supplemental Tables.

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
