# Peer review of "Soil organic nitrogen variation shaped by diverse agroecosystems in a typical karst area: evidence from isotopic geochemistry"

_PeerJ, doi:10.7717/peerj.17221_

## Round 0.1 · original submission · Major Revisions

Although both the reviewers have labelled their decision as 'minor revisions', I think the improvements required according to their comments merit 'major revisions'. The authors are advised to consider all comments carefully.

**Language Note:** PeerJ staff have identified that the English language needs to be improved. When you prepare your next revision, please either (i) have a colleague who is proficient in English and familiar with the subject matter review your manuscript, or (ii) contact a professional editing service to review your manuscript. PeerJ can provide language editing services - you can contact us at [email protected] for pricing (be sure to provide your manuscript number and title). – PeerJ Staff

Reviewer 1 ·

Basic reporting

This article is well written with good English command and is well structured. However, since this article is not the focus of soil erosion, the soil erosion map in Figure 1a is not suitable. In addition, Figure 1 should mark the location of Muhuang Town, or Yinjiang County ;

Experimental design

The information of stratigraphic and lithology of the sampling site should be given. The conclusion that tillage causes the difference of soil pH and element content in AC profile is not sufficient (line362-367), and it is more likely to be caused by the difference of geological background.

Validity of the findings

It is not rigorous to say geological background analog within the mall catchment in Lines212-213;
lines237-238, SON contents continued to decline due to microbial abundance should supplement the literature.

Additional comments

The ' fingerprint ' in the title is not suitable. Apart from the title, the paper does not see the demonstration of the fingerprint of nitrogen isotopes, and it is recommended to delete it.

There are few discussions on the characteristics of soil organic nitrogen in karst areas, and the influence of calcium-rich alkaline on nitrogen turnover should be supplemented in the discussion.

·

Basic reporting

This study conducted sampling on soil profiles of diverse lands uses to investigate SON variations and its response to coupled natural and anthropogenic disturbances in karst regions. Corresponding conclusions are useful for further evaluating soil development and its influence on agricultural management in fragile karst ecosystems.
The “dynamics” in the title seems to be inapposite as a single sampling campaign in this study cannot provide evidence to support discussions over time scales.
If there are some coupled discussions on the correlations between SON and soil organic carbon (SOC), it may be more persuasive to highlight the contribution of this study to the sustainable development of agroecosystems in karst regions.
This study lacks some comparison with other karst regions, e.g., Puding in Guizhou and Huanjiang in Guangxi, to elevate its scientific significance.
Microbial activities showing significant seasonal differences are important controls on both SON and SOC variations, which cannot be well reflected by only a single sampling in Sept. It needs more evidence to support your discussions.

Experimental design

no comment

Validity of the findings

no comment

---

## Round 0.2 · accepted · Accept

The authors have addressed the comments given by the previous reviewers and the said reviewers are satisfied with the revisions done in the manuscript.

Reviewer 1 ·

Basic reporting

no comment

Experimental design

no comment

Validity of the findings

no comment

Additional comments

I have read the revised manuscript carefully, the authors have made the necessary modifications, and the argument is sufficient. It is recommended that the editorial department accept this version for publication.

·

Basic reporting

no comment

Experimental design

no comment

Validity of the findings

no comment

Additional comments

I think the authors have basically addressed the comments or questions.